# Factors Associated with Vaccine Refusal (Polio and Routine Immunization) in High-Risk Areas of Pakistan: A Matched Case-Control Study

**DOI:** 10.3390/vaccines11050947

**Published:** 2023-05-05

**Authors:** Sajid Bashir Soofi, Khadija Vadsaria, Sara Mannan, Muhammad Atif Habib, Farhana Tabassum, Imtiaz Hussain, Sajid Muhammad, Khalid Feroz, Imran Ahmed, Muhammad Islam, Zulfiqar A. Bhutta

**Affiliations:** 1Centre of Excellence in Women and Child Health, Aga Khan University, Karachi 74800, Pakistan; sajid.soofi@aku.edu (S.B.S.);; 2Institute for Global Health and Development, Aga Khan University, Karachi 74800, Pakistan; 3Centre for Global Child Health, The Hospital for Sick Children, 686 Bay Street, Toronto, ON M5G 0A4, Canada

**Keywords:** polio, routine vaccination, refusal, high-risk areas, Pakistan

## Abstract

Background: Pakistan has subpar childhood immunization rates and immunization activities have faced several challenges over the past years. We evaluated the social-behavioral and cultural barriers and risk factors for refusal of polio, Routine Immunization (RI), or both in high-risk areas of poliovirus circulation. Methods: A matched case-control study was conducted from April to July 2017 in eight super high-risk Union Councils of five towns in Karachi, Pakistan. A total of 3 groups, each with 250 cases, including refusals for the Oral Polio Vaccine (OPV) in campaigns (national immunization days and supplementary immunization activities), RI, and both, were matched with 500 controls and identified using surveillance records. Sociodemographic characteristics, household information, and immunization history were assessed. Study outcomes included social-behavioral and cultural barriers and reasons for vaccine refusal. Data were analyzed in STATA using conditional logistic regression. Results: RI refusal was associated with illiteracy and fear of the vaccine’s adverse effects, while OPV refusals were linked to the mother’s decision authority and the assumption that the OPV caused infertility. Conversely, higher socioeconomic status (SES) and knowledge of and willingness to vaccinate with Inactivated Polio Vaccine (IPV) were inversely associated with RI; and lower SES, walking to the vaccination point, knowledge of IPV, and an understanding of contracting polio were inversely associated with OPV refusals, with the latter two also inversely associated with complete vaccine refusal. Conclusion: Education, knowledge and understanding of vaccines, and socioeconomic determinants influenced OPV and RI refusals among children. Effective interventions are needed to address knowledge gaps and misconceptions among parents.

## 1. Introduction

Globally, Pakistan ranks third for being home to the highest number of unvaccinated children (1.2 million), after Nigeria and India, and has the highest neonatal, infant, and child mortality [1]. Immunization, one of the most successful and cost-effective public health interventions, has the potential to save millions of lives annually, thereby preventing the majority of these deaths [1]. Ever since the launch of the Global Polio Eradication Program (GPEI), with the resolution of the World Health Assembly in 1988, tremendous progress has been made toward the global incidence of poliomyelitis, which has been reduced by nearly 99% [2]. Having overcome a situation where the wild-type poliovirus was endemic in 125 countries across five continents, transmission is now limited to two countries—Pakistan and Afghanistan.

The use of these vaccines rapidly interrupted the vicious cycle of poliovirus transmission, which led to the eradication of the wild type two poliovirus in 1999, followed by the last case of the wild type two poliovirus, reported in November 2012 [3]. Routine immunization (RI) against poliovirus and five other diseases was also initiated in 1978 with the launch of the National Expanded Programme on Immunizations (EPI) in Pakistan [4]. However, the introduction and implementation of Supplementary Immunization Activities (SIAs) explicitly targeting polio have been a keystone of the GPEI to increase population immunity [5]. In 2015, four WHO regions were classified as polio-free, leaving Pakistan and Afghanistan as the remaining endemic reservoirs of the wild poliovirus [6].

Despite these efforts and the availability of vaccines, immunization in Pakistan has not yet reached its maximum potential, mainly because its success is contingent on high rates of acceptance and coverage among its people [1].

Evidence from several local studies has demonstrated a low vaccine uptake among children in Pakistan. A community-based survey reported that the Pentavalent and Inactivated Poliovirus Vaccine (IPV) coverage at 14 weeks was limited to 37% and 32% of children in 2017 and 39% and 42% in 2018. In addition, Oral Polio Vaccine (OPV) coverage decreased with each successive vaccination visit, from 66% at birth to 42% at 14 weeks [7]. Low vaccination rates were also highlighted in the Pakistan Demographic and Health Survey (PDHS) 2018 and Third-Party Verification Immunization Coverage Survey (TPVICS) 2021. The PDHS indicated that only 66% of children aged 12 to 23 months had received essential vaccines (one dose of BCG and measles and three doses of DPT and polio vaccines), 51% had all age-appropriate vaccinations, and 4% did not receive any vaccine [8]. According to the latter survey, 76.5% of children under two received complete vaccination, 18.1% partial vaccination, and 5.4% received no vaccines [1]. Suboptimal immunization has also been reported by a number of studies from various areas of Pakistan [9,10].

Similarly, Afghanistan also has a comparable vaccine uptake, where 51.4% of children were reported to be fully vaccinated and 31% were partially vaccinated [11,12].

Low immunization levels are responsible for the high prevalence of vaccine-preventable diseases and the persistence of polio in the high-risk circulation areas of the country. Vaccine uptake in Pakistan is influenced by several individual, social, behavioral, and cultural factors. Factors indicated to enhance vaccination rates include improved parental education and awareness, maternal empowerment, high socioeconomic conditions, and previous visits by health workers [10,13,14]. On the other hand, insufficient belief in immunization, compromised caregiver experiences, vaccine misconceptions, fear of side effects, unavailability of the mother, and distance to the vaccination facility negatively influenced polio and routine immunization [1,15,16,17]. Furthermore, widespread skepticism about vaccine ingredients, as many people believe that vaccines are made of porcine materials—which are strongly forbidden in Islam—is also a significant barrier to vaccine administration and delivery [18]. Similar factors have also been reported in Afghanistan [12].

In addition, geographical limitations, insecurity, and conflict in various areas of Pakistan (Federally Administered Tribal Area (FATA), Khyber Pakhtunkhwa (KP), high-risk areas of Karachi) and Afghanistan, coupled with the polio eradication program’s performance, deep-rooted factors such as community hostility and anger, a fragile health care system, poor public health delivery, a lack of political consensus on the critical nature of eradication, divided and dysfunctional working relationships between government and related stakeholders, and rumors and misinformation from social media and organized boycotts have reduced the acceptance of polio vaccines and attributed to lingering polio cases over the years [19,20,21,22,23]. 

Similarly, common beliefs documented in Pakistan and Afghanistan are that the OPV causes infertility, which manifests in conjunction with the deployment of all-male vaccinator teams in conservative areas and serves as another reason for opposition [24]. Moreover, socio-political setbacks, targeted attacks on polio workers, and the spike in rumors and misinformation about polio vaccination and anti-vaccine conspiracy theories alleging invasion of foreign bodies through polio campaigns to destabilize the country have added to the slew of challenges, reducing people’s intention to vaccinate their children [19,25,26,27].

Improving immunization levels across the country is essential to achieve the targets enacted under the United Nations Sustainable Development Goal 3 to end all preventable deaths among newborns and children under five by 2030 [28].

There are still substantial challenges to overcome before the prospect of a polio-free future for the children of Pakistan can be materialized. Failure to reach sufficient vaccine coverage will mean that polio will continue to cripple children in Pakistan in the future as well.

Limited in-depth analysis of social, behavioral, and cultural factors underlying the OPV and RI refusal exists in the high-risk areas of poliovirus circulation in Karachi. Further, most of the available evidence is survey-based or qualitative, which provides minimal support to establish the causal link between barriers and refusals. Thorough knowledge and comprehension of these challenges within the specific areas are essential to design an appropriate and sustainable strategy to overcome vaccine refusals. Therefore, we undertook a case-control study (nested within an implementation research project in Pakistan) to determine the social-behavioral and cultural reasons and factors for polio non-receipt children during NIDs or SIAs (zero doses for the OPV) and RI non-receipt children. Additionally, our study aimed to assess the barriers related to personal and immunization systems, affecting routine and polio immunization during NIDs or SIAs.

## 2. Material/Methods

### 2.1. Study Design and Setting

We used the matched case-control study design to evaluate social-behavioral, cultural reasons, and system barriers related to the refusal of the OPV, RI, or both. The Aga Khan University conducted the study in Karachi, the provincial capital of Sindh and the major economic hub and the largest city of Pakistan, spread over 3500 square kilometers. Karachi is administratively divided into eighteen towns and Union Councils (UCs). A UC, with a population of approximately 30,000, is the smallest administrative unit in urban and semi-urban areas of Pakistan. The study was conducted from April to July 2017 in eight super high-risk UCs of five towns in Karachi and was based on the highest susceptibility of poliovirus circulation in these areas. These towns were Gadap (UC 4, 5, and 8), Landhi (UC 1 and 2), Site Area (UC 9), Baldia (UC 2), and Orangi (UC 7).

### 2.2. Study Participants

The target population was children aged 0–59 months. The study sample comprised those children who fulfilled the eligibility criteria. Cases were defined as children aged 0 to 59 months residing in the specified eight UCs of the five towns of Karachi. Group A: refused RI (Measles, Pentavalent, PCV, and BCG) but accepted OPV; Group B: refused OPV during national immunization days (NIDs) and SIAs but accepted RI; Group C: refused OPV, IPV, and RI during the first five years of life through EPI and polio campaigns. Controls were defined as children between 0–59 months of age, who lived in the same catchment area as the cases and completed their immunization according to their age.

Each case was matched to two controls based on the area of residence, gender, and age. The age criteria used for matching included the following: if the age of the selected case was less than six months, then the controls were selected within the range of +/− 1 month, and if the age of the case child was above six months the controls were selected within a 3-month age window.

### 2.3. Eligibility Criteria

Cases were included in the study if they were between 0 and 59 months of age, lived in the specified Union Councils, and refused vaccination, and were excluded if parents refused to participate. Controls were included in the study if their age was within +/− 3 months for cases older than 6 months, within +/− 1 month for cases younger than 6 months, if they were fully immunized, and if they resided in the same area as of cases. They were excluded if parents refused to participate.

### 2.4. Identification of Cases and Controls

In the earlier Maternal and Child Care Project (MCCP) phase two study [29], an active surveillance system was implemented, and logs and vaccination record books were maintained with the information of the true number of children in the catchment area and their vaccination status. This case-control study utilized those established sources to identify families who refused to vaccinate their children with the OPV and/or RI within the catchment area. The lists with the refusals were reviewed, and probability proportional to size (PPS) was used to select households. Controls were selected from the same catchment area, where the fourth house on the right and left of the cases’ house was approached. If the household had more than one eligible child, then one child was randomly selected. In the case of an apartment building and the availability of a house number, the data collection team added and subtracted four to the apartment/house number to get the desired apartment/house to be visited. In case the team did not find any eligible child, they followed the same sequence until they enrolled two controls for each case. Recruitment of controls from the same population as of cases and matching on the defined characteristics minimized the selection bias.

### 2.5. Sample Size Estimation

We estimated a sample of 250 cases for each group to detect an odds ratio of 1.8 with 80% power and a 5% significance level. The correlation between cases and controls was assumed as 0.2 as no prior estimate was available. A conservative estimate of the odds ratio was used due to the limited availability of data on factors associated with the lack of OPV coverage. For each case, a matching sample of two controls was selected. Hence, the final sample size of 2250 children (750 cases, 1500 controls) was achieved. The sample size was calculated using NCSS PASS 11 software.

### 2.6. Data Collection

We used a structured questionnaire to collect data from the mothers or caretakers of the study participants. The questionnaire comprised several sections, as described below:

#### 2.6.1. Baseline and Household Information

Information about the age, gender, relationship with the child, years of formal education, profession, number of household members including children under five years, and duration of residence in the area was assessed for the respondents and the household’s head. In addition, the language spoken in the household, details of the health care provider’s visit in the previous month, water access, sanitation, hygiene (WASH), and socioeconomic indicators (house structure, rooms, facilities, availability of electricity, fuel, and possession of assets) were documented.

#### 2.6.2. Immunization History

Details concerning the child’s vaccination status, possession of a vaccination card, and essential vaccines administered at birth (BCG, OPV 0), 6 weeks (Pentavalent 1, PCV 1, OPV 1), 10 weeks (Pentavalent 2, PCV, 2, OPV 2), 14 weeks (Pentavalent 3, PCV 3, OPV 3), 9 months (measles 1, IPV), and 15 months (Measles 2, IPV) were collected.

#### 2.6.3. Knowledge, Attitude, Perception, and Practices of Immunization

A set of questions examined respondents’ knowledge and understanding of immunization in general, and the OPV and IPV in particular, as well as the importance of vaccines in disease prevention and perceptions of vaccinating children. In addition, a comprehensive account of the obstacles, challenges, and reasons for the lack of vaccination and health-seeking behavior was obtained. Furthermore, information about the facilities visited for vaccination, satisfaction with the vaccinators, travel time to the vaccination point, and associated challenges were also evaluated.

The interviews were conducted during the day between 9 am and 4 pm with a parent or caregiver after taking the informed consent. We expected mothers to be present at the time of the visit and in cases of unavailability, a later visit was scheduled. Discussion on the vaccination and vaccine-related activities was kept minimal to avoid bias in exposure assessment.

We recruited data collectors from the local areas with background knowledge of the language and culture of that area. Extensive training was imparted over five days by the senior members of the study team for the conceptual clarity of the methodology and data collection questionnaire. We also conducted a pilot data collection to understand the flow of the questionnaire and any difficulties in conducting the interview.

### 2.7. Data Management and Statistical Analysis

Collected data were meticulously checked on a daily basis by the field supervisors for completeness and accuracy. The completed questionnaires were sent to the Data Management Center at Aga Khan University on a weekly basis for data entry. The data entry database was developed using Visual Basic NET with consistency checks and functions to identify errors and minimize the entry of erroneous data.

After cleaning the data and ensuring quality, we performed the statistical analysis using STATA version 14. Each file was converted from Visual Basic NET to STATA for analysis. Descriptive statistics for cases and controls were reported as frequencies and proportions. For inferential analysis, conditional logistic regression was conducted to ensure adequate statistical power and efficiency. In order to categorize participants into quintiles, socioeconomic status variables were examined using a component analysis. All potential covariates were examined independently in univariate analysis. The covariates with a *p*-value < 0.25 at the univariate level were considered for inclusion in the multivariable model. Results were reported as the matched odds ratios (MOR) with respective 95% CI. The final model was selected based on the predictors’ theoretical and statistical significance. A type I error rate of 0.05 was used for significance.

### 2.8. Ethical Consideration

The Ethics Review Committee of Aga Khan University approved the study (ID # 4876-Ped-ERC-17). All the study participants provided verbal informed consent before the start of the interview. The study team maintained the confidentiality of the participants. All the collected data were stored in a lock, key, and password-protected file, with access granted to only the immediate study team.

## 3. Results

### 3.1. Population and Household Characteristics

The study included 750 cases (250 in each group) and 1500 controls (500 in each group), with a 100% response rate. The population and household characteristics of the study participants are displayed in Table 1. The household density ranged from 9.0–10.6, with a maximum M:F ratio of 1.1. The number of 0–59 months children per household ranged from 1.9 to 2.4, with a M:F sex ratio of 0.9 to 1.2 across the three groups. The most frequently spoken language was Pashto. One-fourth of the households shared a single room for sleeping, and a low proportion used one or more methods to make drinking water safer. On the other hand, a more significant proportion had improved water and toilet facility, used natural gas for cooking, had electricity, and lived in their own house.

### 3.2. Descriptive Analysis of Risk Factors and Barriers across the Three Refusal Groups

Table 2 shows the distribution of risk factors for refusals of RI, OPV, and all vaccinations among each group. For the educational level, illiteracy was prevalent in all three groups but more prevalent among the cases than in the controls (Group A 88.8% vs. 73.8%; Group B 70.8% vs. 66.4%; Group C 83.6% vs. 73.2%). Controls were more educated than cases in all other schooling categories, i.e., primary, middle, secondary, and higher. The knowledge of vaccination and polio disease was high among all the cases, with controls reporting 100% knowledge about immunization. However, IPV knowledge was considerably minimal among cases in Group A (9.2% vs. 77.6%) and C (6.8% vs. 74%) compared to the controls.

Table 3 describes the distribution of barriers related to knowledge, beliefs, and reasons for refusals across the three study groups. Groups A and C, which included the cases of RI and complete vaccine refusal, reported 0% of children receiving RI in Group A and only 2.9 percent in Group C. While Group B, which accepted RI, showed a 98.4% vaccination rate. The most common barriers to RI in these two refusal groups (Groups A and C) were a lack of belief in vaccines’ ability to prevent disease, a lack of awareness and funding, and fear of vaccine-related adverse effects. Groups B and C reported 0% of children receiving the OPV, including refusals for the OPV and complete vaccines. On the other hand, Group A refused RI, but 100% vaccinated their children with the OPV. The common negative perceptions held against the OPV included the vaccine’s capacity to cause infertility (Group B: 56% vs. 54.9%; Group C: 59.5% vs. 54.7%) and its unsafe nature (Group B: 26.9% vs. 25.2%; Group C: 27.8% vs. 26.0%). Additionally, the most often cited reasons for refusals were concern about the negative impact (Group B: 25.2 vs. 0%; Group C 14.5% vs. 7.4), lack of benefits of the OPV (Group B: 17.5% vs. 0%; Group C: 23.3% vs. 0%), and a dislike of polio workers visiting the household (Group B: 8.1% vs. 5.3%; Group C: 8.8% vs. 0%). On the other hand, the IPV was given to 100% of children in Group B and 21.7% in Group A, while no case child was vaccinated with the IPV in group C. The most common reasons for IPV refusals included the child receiving multiple OPV doses (Group A: 16.7% vs. 7%), family refusal (Group A 44.4% vs. 4%; Group C 17.7% vs. 0%), vaccine not halal (Group A 11.1% vs. 0%), unsafe (Group A 22.2% vs. 0%; Group C 29.4% vs. 0%), and can cause infertility (Group A 11.1% vs. 0%; Group C 17.7% vs. 0%). In addition, a mother needing permission to vaccinate their child was the common barrier reported across all three groups (Group A: 78.4% vs. 60.8%, Group B 74%% vs. 65%%; Group C 74% vs. 64%), with a majority of the mothers needing permission from the husband to vaccinate.

### 3.3. Inferential Analysis of Risk Factors and Barriers to RI, OPV, and Both

Each refusal group was assessed using univariate and multivariable analysis for its associated factors. The number of cases in each group represents the true refusals (Group A- 55.2%; Group B- 44%, and Group C- 27.6%).

#### 3.3.1. Group A: Refusal for Routine Immunization (RI) but Acceptance of OPV

Cases were significantly likely to refuse RI compared to controls due to illiteracy (OR 4.13; CI 1.97–8.64), mother owning the authority to make vaccination decision (OR 2.10; CI 1.29–3.41), receiving multiple OPV doses (OR 76; CI 10.43–553.53), considering the OPV unsafe (OR 64; CI 8.74–468.36), and fearing the adverse effects of the vaccine (OR 2.22; CI 1.43–3.47). On the other hand, factors such as cases being rich (OR 0.37; CI 0.17–0.84), knowledge of polio prevention through vaccination (OR 0.05; CI 0.02–0.09), possessing adequate knowledge about OPV doses (OR 0.05; CI 0.02–0.15) and the IPV (OR 0.01; CI 0.003–0.04), understanding the need of the OPV after the IPV (OR 0.01: CI 0.003–0.05), considering vaccine halal (OR 0.56; CI 0.29–1.06) and safe (OR 0.12; CI 0.07–0.21), showing a willingness to vaccinate with the IPV (OR 0.03; CI 0.004–0.20), and understanding the child’s risk of acquiring polio (OR 0.40; CI 0.18–0.87) were inversely associated with the refusals for RI. These factors were assessed in the multivariable model, which also revealed that illiteracy (AOR 3.95; CI 1.85–8.39) and fear of adverse effects of the vaccine (AOR 6.05; CI 1.07–33.57) as the most frequent reasons for RI refusals, whereas the refusals were inversely related to being rich (AOR: 0.36; CI 0.16–0.82), possessing knowledge about the IPV (AOR 0.01; CI 0.0003–0.08), understanding the need of the OPV after the IPV (AOR 0.07; CI 0.01–0.70), along with the willingness to vaccinate with the IPV (AOR 0.01; CI 0.0004; 0.31) (Table 4).

#### 3.3.2. Group B: Refusal of OPV (NIDs and SIAs) but Acceptance of RI

OPV refusals were significantly associated with the mother’s authority to decide about the vaccination (OR 1.75; CI 0.95–3.19) and whether they considered the OPV to cause infertility (OR 2.74; CI 1.71–4.39). In contrast, cases were less likely to refuse the OPV if they were poor (OR 0.32; CI 0.12–0.86), had knowledge about polio disease prevention through vaccination (OR 0.55; CI 0.31–0.97), had knowledge about the IPV (OR 0.27; CI 0.13–0.55), understood the need of the OPV after the IPV (OR 0.06; CI: 0.02–0.14), considered the vaccine to be halal (OR 0.24; CI 0.09–0.65) and safe (OR 0.01; CI 0.001–0.05), showed a willingness to vaccinate with the IPV (OR 0.29; CI 0.12–0.68), walked to the vaccination point (OR 0.54; CI 0.33–0.87), were satisfied with the performance of the vaccinators (OR 0.25; CI 0.08–0.73), and understood the child’s risk of acquiring polio (OR 0.16; CI 0.07–0.36). The barriers to OPV vaccination were also significant at the multivariable level (mother holding authority to decide (OR 2.96; CI 1.01–8.7) and fear of vaccine causing infertility (OR 2.49; CI 1.15–5.42)). Whereas, being poor (OR 0.32; CI 0.12–0.86), possessing knowledge about IPV (OR 0.29; CI 0.10–0.90), understanding the need for the OPV after the IPV (OR 0.03; CI 0.01–0.11), and the risk associated with not vaccinating with the OPV (OR 0.03; CI 0.01–0.13), and walking to the vaccination point (OR 0.40; CI 0.17–0.94) were the factors inversely associated with the refusal of the OPV (Table 5).

#### 3.3.3. Group C: Complete Refusal (RI and OPV)

The barriers such as the belief that the vaccine can cause infertility (OR 2.23; CI 1.16–4.29), the mother’s authority to decide on the vaccination (OR 2.24; CI 1.01–4.94), and considering the IPV unsafe (OR 52; CI 7.06–383.19) were significantly associated with complete vaccine refusal. However, these factors were not significant in the multivariable model. On the other hand, mothers having knowledge about polio prevention through vaccination (AOR 0.07; CI 0.03–0.19) and understanding a child’s risk of acquiring polio (AOR 0.08; CI 0.02–0.32) were inversely associated with vaccine refusal in the univariate as well as multivariable analysis (Table 6).

## 4. Discussion

Suboptimal vaccine uptake and increased refusal of routine immunization and polio vaccines result from a complex interaction between social, behavioral, and cultural factors and logistic barriers prevailing in the eight high-risk UCs of Karachi.

Our study demonstrated that education substantially affects mothers’/families’ acceptance of RI for their children under five. We found that a greater proportion of RI refusal cases (91.3%) were illiterate compared to the controls (76%). Prior research in Pakistan and elsewhere have similarly linked education to vaccination response, showing fewer vaccine refusals with higher parental education levels [10,30,31]. The role of education in impacting health literacy is well established, enabling mothers to make more informed health decisions. Individuals with a higher level of education are more likely to understand the risk of vaccine-preventable diseases, the effectiveness of vaccines, the use of selective information sources, dependence on critical thinking, and the importance of making more proactive decisions [31]. The finding that only 8.7% of cases in the RI refusal group were literate is concerning because it could play a significant role in the outbreak of vaccine-preventable diseases in Pakistan, which would be detrimental to the country and the world.

One in ten children globally lack access to immunization, leaving behind many “zero-dozers”. In contrast, despite having access, many children are not vaccinated due to prevalent misconceptions among parents. In our study, fear of vaccines’ adverse effects was associated with a 6.05-fold increase in the likelihood of RI refusal among cases. Refusal due to adverse effects has also been reported previously, with possible reasons including pain, discomfort, and fever following vaccination, misleading or incorrect information from mass media, unfavorable incidents experienced by friends or family, and a lack of knowledge regarding the role of vaccines in disease prevention [32]. These reasons, coupled with false assumptions and doubts surrounding vaccines, can cause fear and worry among parents. Earlier research has also indicated the reluctance to vaccinate due to apprehension of side effects and concerns about the safety of vaccines as one of the primary reasons for refusals among parents [32,33,34].

In patriarchal societies, usually, males and family elders own decision-making authority. Over half of the respondent mothers in each refusal group reported needing permission from another family member (husband, mother-in-law, brother-in-law, or father-in-law) for their child’s vaccination, reflecting the existing social and familial structures and lack of empowerment, hindering the routine immunization in targeted regions. On the other hand, it is assumed that if mothers had the authority to make health decisions, they would decide in the best interest of their children. This assumption was further supported by a study that analyzed data from demographic and health surveys of Bangladesh, Cambodia, Indonesia, Nepal, Pakistan, and the Philippines between 2011 and 2014 and found that children of mothers who had more decisional authority were more likely to be vaccinated than those who did not [35]. However, we found contrasting evidence, where mothers with decision-making authority in our study reported more refusals for OPV.

This observation reveals several widespread myths, misunderstandings, and lack of knowledge that give rise to frequent campaigns that the OPV is used as a birth control strategy, that vaccines contain substances forbidden by Islam, and a suspicion that the OPV is a foreign conspiracy and an artificial alteration of the fate determined by God, especially when the disease has not been experienced by vaccine recipient [36]. A study in Malaysia reported that 76% of mothers who refused childhood vaccination perceived vaccine ingredients not to be halal, and 78% refused vaccines due to their religious beliefs [37].

Except for the belief that the OPV/IPV can induce infertility, none of the factors cited in our study for OPV refusal were statistically significant. A study conducted in Northern Nigeria reported that 30% of respondents were concerned about the vaccine’s safety, and more than half were concerned that it would cause infertility in their children [38]. Due to misinformation and illiteracy, these misconceptions will likely prevail and spread, resulting in mothers’ reluctance to accept the OPV/IPV in campaigns. The mothers’ knowledge is considered a significant component in influencing their decisions regarding the vaccination of children. (7) [39]. It is also emphasized that instead of only the mother owning the decision authority, shared decision making among parents is associated with less chance of refusals [31].

Our study also revealed a number of factors that were inversely associated with refusals across all three groups. We found that mothers/families who were rich were less likely to refuse RI; similar notions have been reported in earlier research [40]. Literacy is influenced by numerous socioeconomic factors, with poverty being the critical reason believed to have coexisted with illiteracy and low accomplishment. In developing countries, a definite association exists between poverty and illiteracy, as the former forces individuals to engage in child labor, leading to illiteracy, and these illiterate individuals take on low-income jobs, exacerbating poverty [41]. Consequently, those who have higher SES are more likely to be literate and less likely to refuse RI than their counterparts. Literacy is also linked with knowledge and awareness, where mothers/families with a greater understanding of the purpose of polio vaccination were less likely to refuse RI and the OPV/IPV. In turn, these mothers were more willing to vaccinate their children with the IPV.

Vaccination efforts constitute Pakistan’s backbone in its fight against polio, and door-to-door campaigns are a vital part of SIAs. Due to the fact that a child must receive multiple doses to build an adequate immune response, Pakistan conducts multiple campaigns every year, with a greater number of campaigns in regions with the highest risk of transmission. Evidence suggests that people are more likely to vaccinate their children if they believe others in their social group do the same or if their religious beliefs match those of others. Around half of the cases in the OPV refusal group belonged to the poorest to middle SES. These mothers were more likely to accept the OPV if they perceived their neighbors/social groups accepted it during the campaigns, encouraging them to vaccinate their children. This logic may be particularly applicable in Pakistan, where the vaccination experience is socially mediated [42].

Furthermore, our study also revealed an interesting finding that mothers were less likely to refuse the OPV when they walked to the vaccination site. Earlier research has reported contrasting results when comparing the distance to the health facility and vaccine uptake. Our findings revealed that many respondents lived up to 30 min from the vaccination center, and over half reached the vaccination point via walking. In a study carried out in Pakistan, 18% of children with <3 routine OPV doses reported vaccination posts to be too far away [43]. Similar results were stated by another study that found children who lived further from the immunization facility were less likely to be vaccinated [44].

In contrast, a study showed that parents were prepared to make considerable effort to have their children immunized [40]. The time and physical effort required to reach vaccination sites for multiple campaigns and the burden of traveling on foot with one or more children could potentially deter people from seeking and returning for vaccination services. In our situation, walking to the immunization site reduced OPV refusal. One of the reasons reported by mothers is a dislike toward frequent visits of vaccinators to their houses. Another explanation points toward the fact that mothers willing to walk to the site took it as an opportunity to socialize with other mothers in their neighborhood, which otherwise is less likely due to the conservative social structures.

In order to address vaccine refusals and reduce the “zero dozers”, parents must be educated to overcome misconceptions, false beliefs, and concerns related to vaccines. In addition, parents who immunize their children should be reinforced so as to prevent them from experiencing weariness. Numerous studies have shown that educating mothers with limited or no formal education on the relevance of vaccines for child health can substantially increase the vaccination rate [40]. It is essential to understand that immunizing children may not be a priority for mothers or caregivers who are overburdened with household or work responsibilities. As a result, disadvantaged children may encounter several logistical challenges when trying to access vaccination services, including a lack of resources or transportation, mothers’ inability to get to the vaccination site during the campaign days, a lack of vaccines at the site, or a conflict between the vaccination hours and parents’ work schedules [31].

## 5. Strengths and Limitations

Our study comprehensively analyzed the barriers and risk factors associated with refusals across the three categories. It is one of the few studies to investigate the outcomes in high-risk areas of Karachi, where poliovirus circulation continues to impede eradication efforts. Moreover, we employed a case-control design, which is superior to surveys in generating scientific evidence and can be used by relevant programs and policymakers to plan and implement strategies to address the identified barriers. A limitation of the case-control study design is selection bias, which the authors overcame by selecting the controls from the same source population as the cases. In addition, age, gender, and area of residence were used to match controls with cases to assure equal distributions of covariates that can influence exposure and improve efficiency. Moreover, even though we present the 2017 data analysis, our findings are still applicable and useful for programmes and policies that address the infrequently targeted barriers and obstacles in high-risk regions.

## 6. Conclusions

In conclusion, immunization is one of the most cost-effective public health interventions, with considerable impact on health. Our study highlighted that the illiteracy and fear of the vaccine’s adverse effects were the significant barriers to RI, whereas the mother’s decision authority and belief that the vaccine causes infertility were the significant barriers to the OPV. Further, our study also indicated the protective role of socioeconomic status, adequate knowledge and understanding of polio vaccines, willingness to vaccinate a child with IPV, understanding of the vaccine’s role in disease prevention, and acknowledging the risk of contracting polio with vaccine refusals. Understanding these risk factors is necessary for developing a comprehensive strategy to target areas with zero-dozer children and children with incomplete immunization to finally eliminate these obstacles. To bring Pakistan one step closer to polio eradication and attain the vaccination coverage benchmark, a participatory approach should be considered when designing a tailored intervention where parents and care givers can become active agents of the desired change.

## Figures and Tables

**Table 1 vaccines-11-00947-t001:** Population and household characteristics.

Population and HH Characteristics	Group A *		Group B *		Group C *	
	Cases (*n* = 250)	Control (*n* = 500)	Cases (*n* = 250)	Control (*n* = 500)	Cases (*n* = 250)	Control (*n* = 500)
Households (*n*)	250	500	250	500	250	500
Total population (*n*)	2497	4739	2241	4800	2641	4950
Household density (*n*)	10.0	9.5	9.0	9.6	10.6	9.9
Sex ratio all members (M:F)	1.0	1.0	1.0	1.0	1.1	1.1
Children 0–59 months per household	2.2	2.2	1.9	2.2	2.4	2.3
Sex ratio for children 0–59 months (M:F)	0.9	1.1	1.0	0.9	1.2	1.2
**Language Spoken in household *n* (%)**						
Urdu	8 (3.2)	42 (8.4)	24 (9.6)	46 (9.2)	6 (2.4)	24 (4.8)
Sindhi	19 (7.6)	13 (2.6)	4 (1.6)	17 (3.4)	3 (1.2)	20 (4.0)
Punjabi	9 (3.6)	27 (5.4)	4 (1.6)	29 (5.8)	4 (1.6)	22 (4.4)
Pashto	171 (68.4)	320 (64.0)	169 (67.6)	300 (60.0)	198 (79.2)	342 (68.4)
Balochi	7 (2.8)	12 (2.4)	1 (0.4)	6 (1.2)	2 (0.8)	9 (1.8)
Seraiki	11 (4.4)	15 (3.0)	0 (0.0)	16 (3.2)	5 (2.0)	8 (1.6)
Others	25 (10.0)	71 (14.2)	48 (19.2)	86 (17.2)	32 (12.8)	75 (15.0)
**Proportion of households *n* (%)**						
Using single room for sleeping	68 (27.2)	122 (24.4)	46 (18.4)	101 (20.2)	55 (22.0)	116 (23.2)
With improved water facility ^~^	200 (80.0)	407 (81.4)	193 (77.2)	397 (79.4)	195 (78.0)	399 (79.8)
Using one or more method for making water safer to drink	28 (11.2)	95 (19.0)	43 (17.2)	99 (19.8)	45 (18.0)	102 (20.4)
With improved toilet facility ‡	242 (96.8)	473 (94.6)	244 (97.6)	472 (94.4)	233 (93.2)	484 (96.8)
Sharing toilet facility with others	3 (1.2)	1 (0.2)	0 (0.0)	3 (0.6)	0 (0.0)	6 (1.2)
Using natural gas for cooking	222 (88.8)	414 (82.8)	224 (89.6)	436 (87.2)	216 (86.4)	430 (86.0)
Having electricity	244 (97.6)	499 (99.8)	249 (99.6)	500 (100.0)	250 (100.0)	499 (99.8)
Living in owned house	139 (55.6)	330 (66.0)	181 (72.4)	336 (67.2)	168 (67.2)	332 (66.4)

* Group A: refusal of routine immunization (RI) but acceptance of OPV. * Group B: refusal of OPV (NIDs and SIAs) but acceptance of RI. * Group C: complete vaccine refusals; routine and campaigns. ^~^ Including piped into dwelling, public tap/standpipe, tube well or borehole, hand pump, protected well, protected spring, rainwater, filtration plant, and bottled water. ‡ Including flush to piped sewer system, flush to septic tank, and flush to pit latrine.

**Table 2 vaccines-11-00947-t002:** Descriptive analysis of the risk factors for refusal.

Risk Factors for Refusals	Group A *		Group B *		Group C *	
	Cases *n* = 250 *n* (%)	Controls *n* = 500 *n* (%)	Cases *n* = 250 *n* (%)	Controls *n* = 500 *n* (%)	Cases *n* = 250 *n* (%)	Controls *n* = 500 *n* (%)
**Education level**						
Illiterate	222 (88.8)	369 (73.8)	177 (70.8)	332 (66.4)	209 (83.6)	366 (73.2)
Primary and middle school	24 (9.6)	86 (17.2)	41 (16.4)	100 (20.0)	30 (12)	85 (17.0)
Secondary school	4 (1.6)	31 (6.2)	22 (8.8)	52 (10.4)	8 (3.2)	32 (6.4)
Higher secondary school and above	0 (0)	14 (2.8)	10 (4)	16 (3.2)	3 (1.2)	17 (3.4)
**Knowledge level**						
Knowledge about vaccination	247 (98.8)	500 (100.0)	250 (100)	500 (100.0)	245 (98)	500 (100.0)
Knowledge about polio disease	247 (98.8)	497 (99.4)	245 (98)	497 (99.4)	240 (96)	497 (99.4)
Knowledge about IPV	23 (9.2)	388 (77.6)	157 (62.8)	397 (79.4)	17 (6.8)	370 (74.0)

* Group A: refusal of routine immunization (RI) but acceptance of OPV. * Group B: refusal of OPV (NIDs and SIAs) but acceptance of RI. * Group C: complete vaccine refusals; routine and campaigns.

**Table 3 vaccines-11-00947-t003:** Descriptive analysis of barriers associated with refusal.

Risk Factors for Refusals	Group A *		Group B *		Group C *	
	Cases *n* = 250 *n* (%)	Controls *n* = 500 *n* (%)	Cases *n* = 250 *n* (%)	Controls *n* = 500 *n* (%)	Cases *n* = 250 *n* (%)	Controls *n* = 500 *n* (%)
**Barriers to Vaccination/RI**						
Parents vaccinate their child	0 (0.0)	500 (100)	246 (98.4)	500 (100)	7 (2.9)	500 (100)
Belief that vaccine prevents diseases	108 (43.7)	497 (99.4)	241 (96.4)	495 (99.0)	93 (38)	495 (99.0)
Lack of education	5 (2)	40 (8.0)	6 (2.4)	33 (6.6)	3 (1.2)	31 (6.2)
Lack of funds	7 (2.8)	25 (5.0)	11 (4.4)	21 (4.2)	6 (2.5)	33 (6.6)
Lack of awareness	44 (17.8)	56 (11.2)	28 (11.2)	48 (9.6)	34 (13.9)	49 (9.8)
Lack of facilities	23 (9.3)	31 (6.2)	15 (6)	29 (5.8)	10 (4.1)	28 (5.6)
Fear of adverse effects of vaccine	116 (47)	202 (40.4)	93 (37.2)	209 (41.8)	142 (58)	213 (42.6)
**Barriers with OPV**						
Parent gives OPV to child	250 (100)	500 (100.0)	0 (0)	500 (100.0)	0 (0)	500 (100.0)
** *Negative perceptions of OPV:* **						
Heard that vaccine is not halal	20 (9.9)	75 (17.8)	37 (15.8)	78 (18.5)	28 (12.3)	82 (18.7)
Heard that vaccine can cause infertility	118 (58.4)	233 (55.2)	131 (56)	231 (54.9)	135 (59.5)	240 (54.7)
Heard that vaccine is not safe	64 (31.7)	110 (26.1)	63 (26.9)	106 (25.2)	63 (27.8)	114 (26.0)
** *Reasons to not vaccinate child with OPV* ** *:*						
No need to give OPV every campaign	7 (9.9)	2 (10)	1 (0.4)	5 (26.3)	7 (2.8)	4 (14.8)
Worried about the negative impact of OPV	4 (5.6)	0 (0)	62 (25.2)	0 (0.0)	36 (14.5)	2 (7.4)
Not available in every polio campaign	2 (2.8)	0 (0)	1 (0.4)	0 (0.0)	0 (0)	0 (0)
Polio drops are not beneficial	2 (2.8)	1 (5)	43 (17.5)	0 (0.0)	58 (23.3)	0 (0)
Parent doesn’t like the polio worker visiting home	5 (7)	1 (5)	20 (8.1)	1 (5.3)	22 (8.8)	0 (0)
**Barriers with IPV**						
Child received IPV	5 (21.7)	361 (93.0)	157 (100)	387 (97.5)	0 (0)	350 (94.6)
** *Reasons for refusal:* **						
Child has received polio drops too many times	3 (16.7)	2 (7)	-	-	1 (5.9)	2 (10)
Family does not allow for vaccine	8 (44.4)	1 (4)	-	-	3 (17.7)	0 (0)
Vaccine is not halal	2 (11.1)	0 (0)	-	-	0 (0)	0 (0)
Vaccine can cause sterility	2 (11.1)	0 (0)	-	-	3 (17.7)	0 (0)
Vaccine is not safe	4 (22.2)	0 (0)	-	-	5 (29.4)	0 (0)
Against religious belief	0 (0)	0 (0)	-	-	1 (5.9)	0 (0)
**Other barriers**						
Mother needs permission for child’s vaccination	196 (78.4)	304 (60.8)	185 (74)	325 (65.0)	185 (74)	320 (64.0)
Permission from husband	176 (89.8)	271 (89.1)	157 (84.9)	287 (88.3)	147 (79.5)	288 (90.0)
Permission from mother-in-law	9 (4.6)	22 (7.2)	15 (8.1)	24 (7.4)	15 (8.1)	19 (5.9)
Permission from brother-in-law	0 (0)	0 (0)	0 (0)	1 (0.3)	0 (0)	2 (0.6)
Permission from father-in-law	10 (5.1)	11 (3.6)	10 (5.4)	12 (3.7)	20 (10.8)	11 (3.4)

* Group A: refusal of routine immunization (RI) but acceptance of OPV. * Group B: refusal of OPV (NIDs and SIAs) but acceptance of RI. * Group C: complete vaccine refusals; routine and campaigns.

**Table 4 vaccines-11-00947-t004:** Multivariable analysis of risk factors for routine immunization refusals.

	Group A: Refusal of Routine Immunization (RI) but Acceptance of OPV					
Risk Factors	Cases *n* = 138 *n* (%)	Controls *n* = 276 *n* (%)	Unadjusted Matched Odds Ratio (95% CI)	*p*-Value	Adjusted Matched Odds Ratio (95% CI)	*p*-Value *
**Education level:**						
Illiterate	126 (91.3)	210 (76.1)	4.13 (1.97–8.64)	<0.001	3.95 (1.85–8.39)	<0.001
Literate	12 (8.7)	66 (23.9)	Ref		Ref	
**SES Level:**						
Poorest	38 (27.5)	74 (26.8)	0.97 (0.50–1.88)	0.926	0.76 (0.38–1.51)	0.431
Poor	34 (24.6)	60 (21.7)	1.07 (0.56–2.04)	0.831	0.84 (0.43–1.66)	0.624
Middle	25 (18.1)	36 (13.0)	1.26 (0.63–2.50)	0.515	1.09 (0.53–2.27)	0.807
Rich	14 (10.1)	58 (21.0)	0.37 (0.17–0.84)	0.017	0.36 (0.16–0.82)	0.016
Richest	27 (19.6)	48 (17.4)	Ref		Ref	
**Knowledge level:**						
Knowledge about disease prevention through vaccination	0	274 (99.3)	-		-	
Knowledge about polio disease prevention through vaccination	11 (8.0)	206 (74.6)	0.05 (0.02–0.09)	<0.001	-	
Adequate knowledge regarding OPV doses	97 (70.3)	268 (97.1)	0.05 (0.02–0.15)	<0.001	-	
Knowledge about IPV	13 (9.4)	232 (84.1)	0.01 (0.003–0.04)	<0.001	0.01 (0.0003–0.08)	<0.001
Need of OPV after IPV	2 (1.4)	178 (64.5)	0.01 (0.003–0.05)	0.003	0.07 (0.01–0.70)	0.024
Authority of mother in child’s vaccination decisions	102 (73.9)	164 (59.4)	2.10 (1.29–3.41)	<0.001	-	
**Negative news about OPV:**						
Vaccine is not halal	16 (11.6)	50 (18.1)	0.56 (0.29–1.06)	0.075	-	
Vaccine can cause infertility	70 (50.7)	119 (43.1)	1.38 (0.90–2.10)	0.135	-	
Vaccine is not safe	32 (23.2)	64 (23.2)	1.00 (0.62–1.62)	0.999	-	
**Perception about safe OPV vaccine:**						
Safe	40 (29.0)	206 (74.6)	0.12 (0.07–0.21)	<0.001	-	
Unsafe	98 (71.0)	70 (25.4)	Ref	-	-	
Willingness to vaccinate with IPV	4 (2.9)	56 (20.3)	0.03 (0.004–0.20)	<0.001	0.01 (0.0004; 0.31)	0.008
Child has received polio drops too many times	38 (27.5)	1 (0.4)	76 (10.43–553.53)	<0.001	-	
Vaccine is not safe	32 (23.2)	1 (0.4)	64 (8.74–468.36)	<0.001	-	
Fear of adverse effects of vaccine	77 (55.8)	104 (37.7)	2.22 (1.43–3.47)	<0.001	6.05 (1.07–33.57)	0.041
Distance of vaccination center/services: ≤30 min	0	249 (90.2)	-	-	-	
Walking to reach vaccination point	0	158 (57.3)	-	-	-	
Satisfaction with the performance of the vaccinators	0	272 (98.6)	-	-	-	
Parental understanding of child’s risk of acquiring polio	22 (15.9)	64 (23.2)	0.40 (0.18–0.87)	0.022	-	

* Significant *p*-Value < 0.05; CI: Confidence Interval.

**Table 5 vaccines-11-00947-t005:** Multivariable analysis of risk factors among Oral Polio Vaccine (OPV) refusals in campaigns (NIDs and SIAs).

	Group B: Refusal of OPV (NIDs and SIAs) but Acceptance of RI					
Risk Factors	Cases *n* = 138 *n* (%)	Controls *n* = 276 *n* (%)	Unadjusted Matched Odds Ratio (95% CI)	*p*-Value	Adjusted Matched Odds Ratio (95% CI)	*p*-Value *
**Education level:**						
Illiterate	78 (70.9)	151 (68.6)	1.14 (0.66–1.96)	0.645	-	-
Literate	32 (29.1)	69 (31.4)	Ref			
**SES Level:**						
Poorest	10 (9.1)	37 (16.8)	0.32 (0.12–0.86)	0.024	0.32 (0.12–0.86)	0.024
Poor	16 (14.6)	35 (15.9)	0.65 (0.28–1.52)	0.324	0.65 (0.28–1.52)	0.324
Middle	30 (27.3)	47 (21.4)	0.90 (0.43–1.89)	0.79	0.90 (0.43–1.89)	0.79
Rich	28 (25.5)	61 (27.7)	0.68 (0.32–1.41)	0.3	0.68 (0.32–1.41)	0.3
Richest	26 (23.6)	40 (18.2)	Ref	-	Ref	-
**Knowledge level:**						
Knowledge about disease prevention through vaccination	107 (97.3)	216 (98.2)	0.67 (0.15–2.98)	0.596	-	-
Knowledge about polio disease prevention through vaccination	69 (62.7)	160 (72.7)	0.55 (0.31–0.97)	0.038	-	-
Adequate knowledge regarding OPV doses	0	212 (96.4)	-	-	-	-
Knowledge about IPV	76 (69.1)	185 (84.1)	0.27 (0.13–0.55)	<0.001	0.29 (0.10–0.90)	0.031
Need of OPV after IPV	6 (5.4)	120 (54.6)	0.06 (0.02–0.14)	<0.001	0.03 (0.01–0.11)	<0.001
Authority of mother in child’s vaccination decisions	86 (78.2)	153 (69.6)	1.75 (0.95–3.19)	0.07	2.96 (1.01–8.7)	0.049
**Negative news about OPV:**						
Vaccine is not halal	6 (5.4)	37 (16.8)	0.24 (0.09–0.65)	0.005	-	-
Vaccine can cause infertility	76 (69.1)	94 (42.7)	2.74 (1.71–4.39)	<0.001	2.49 (1.15–5.42)	0.021
Vaccine is not safe	27 (24.6)	50 (22.7)	1.10 (0.65–1.87)	0.717	-	-
**Perception about safe OPV vaccine:**						
Safe	3 (2.7)	159 (72.3)	0.01 (0.001–0.05)	<0.001	-	-
Unsafe	107 (97.3)	61 (27.7)	Ref	-	-	-
Willingness to vaccinate with IPV	8 (7.3)	40 (18.2)	0.29 (0.12–0.68)	0.005	-	-
Child has received polio drops too many times	11 (10.0)	0	-	-	-	-
Vaccine is not safe	10 (9.1)	0	-	-	-	-
Fear of adverse effects of vaccine	35 (31.8)	89 (40.4)	0.64 (0.37–1.08)	0.095	-	-
Distance of vaccination center/services: ≤30 min	98 (89.1)	212 (96.4)	0.25 (0.09–0.72)	0.086	-	-
Walking to reach vaccination point	58 (52.7)	147 (66.8)	0.54 (0.33–0.87)	0.012	0.40 (0.17–0.94)	0.036
Satisfaction with the performance of the vaccinators	100 (90.9)	215 (97.7)	0.25 (0.08–0.73)	0.011	-	-
Parental understanding regarding child’s risk of acquiring polio	11 (10.0)	69 (31.4)	0.16 (0.07–0.36)	<0.001	0.03 (0.01–0.13)	<0.001

* Significant *p*-Value < 0.05; CI: Confidence Interval.

**Table 6 vaccines-11-00947-t006:** Multivariable analysis of risk factors for complete vaccine refusals; routine and campaigns.

	Group C: Complete Vaccine Refusals (Routine and Campaigns)					
Risk Factors	Cases *n* = 138 *n* (%)	Controls *n* = 276 *n* (%)	Unadjusted Matched Odds Ratio (95% CI)	*p*-Value	Adjusted Matched Odds Ratio (95% CI)	*p*-Value *
**Education level:**						
Illiterate	57 (82.6)	105 (76.1)	1.6 (0.72–3.55)	0.248	-	
Literate	12 (17.4)	33 (23.9)	Ref.			
**SES Level:**						
Poorest	11 (15.9)	26 (18.8)	0.54 (0.15–1.57)	0.258	-	
Poor	11 (15.9)	14 (10.1)	0.96 (0.34–2.73)	0.938	-	
Middle	14 (20.3)	33 (23.9)	0.55 (0.22–1.34)	0.186	-	
Rich	13 (18.8)	39 (28.3)	0.42 (0.17–1.04)	0.060	-	
Richest	20 (29.0)	26 (18.8)	Ref.		-	
**Knowledge level:**						
Knowledge about disease prevention through vaccination	0	134 (97.1)	-		-	
Knowledge about polio disease prevention through vaccination	7 (10.1)	90 (65.2)	0.09 (0.04–0.20)	<0.001	0.07 (0.03–0.19)	<0.001
Adequate knowledge regarding OPV doses	0	133 (96.4)	-		-	
Knowledge about IPV	4 (5.8)	104 (75.4)	-		-	
Need of OPV after IPV	1 (1.4)	82 (59.4)	-		-	
Authority of mother in child’s vaccination decisions	58 (84.1)	99 (71.7)	2.24 (1.01–4.94)	0.046	-	
**Negative news about OPV:**						
Vaccine is not halal	2 (2.9)	14 (10.1)	0.25 (0.05–1.16)	0.077	-	
Vaccine can cause infertility	46 (66.7)	68 (49.3)	2.23 (1.16–4.29)	0.016	-	
Vaccine is not safe	21 (30.4)	34 (24.6)	1.35 (0.70–2.59)	0.366	-	
**Perception about safe OPV vaccine:**						
Safe	0	97 (70.3)	-	-	-	-
Unsafe	69 (00.0)	41 (29.7)	-	-	-	-
Willingness to vaccinate with IPV	0	39 (28.3)	-	-	-	-
If no, reason for not accepting IPV: child has received polio drops too many times	21 (30.4)	0	-	-	-	-
If no, reason for not accepting IPV: vaccine is not safe	26 (37.7)	1 (0.7)	52 (7.06–383.19	<0.001	-	-
Fear of adverse effects of vaccine	37 (53.6)	59 (42.8)	1.66 (0.88–3.11)	0.116	-	-
Distance of vaccination center/services: ≤30 min	0	128 (92.8)	-		-	-
Walking to reach vaccination point	0	91 (65.9)	-		-	-
Satisfaction with the performance of the vaccinators	0	134 (97.1)	-		-	-
Parental understanding regarding child’s risk of acquiring polio	6 (8.7)	46 (33.3)	0.10 (0.03–0.32)	<0.001	0.08 (0.02–0.32)	<0.001

* Significant *p*-Value < 0.05; CI: Confidence Interval.

## Data Availability

The data presented in this study are available on request from the corresponding author.

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
