# Peer review of "Factors Associated with Vaccine Refusal (Polio and Routine Immunization) in High-Risk Areas of Pakistan: A Matched Case-Control Study"

_vaccines, 2023, doi:10.3390/vaccines11050947_

Round 1

Reviewer 1 Report

Abstract

·         Please provide further information about the study outcomes

·         In the abstract you need to answer the following questions, what, why and how and discuss the study new findings, limitations, and future research

·         The abstract should state briefly the purpose of the research, the principal results and major conclusions. An abstract is often presented separately from the article, so it must be able to stand alone

Introduction

-          discuss the research aims, research gap and discuss the paper layout Add up-to-date references to support your discussion

-          The necessity and innovation of the article should be presented to the introduction

Methods and Materials 

·        The methodology of this study should be detailed, limit information was provided on method and materials

·         The authors should be able to control/reduce  the selection bias

·                Abbreviations should be defined

Discussion

-          I believe that more in depth discussion is needed. The discussion as present now is simple and concise. Revision of more papers using similar technique is needed

-          In the discussion, please discuss if the study research questions are answered or not Also introduce the model in detail. Draw a conclusion from this study and present the limitations and future research.

Minor editing of English language required  in this paper.

Author Response

Abstract

Point 1: Please provide further information about the study outcomes

Response: a statement mentioning the study outcomes has been added separately. The barriers and reasons cannot be elaborated on in detail due to the word count limit for the abstract.

Point 2: In the abstract you need to answer the following questions, what, why and how and discuss the study new findings, limitations, and future research

Response: The abstract has been revised in all parts to address the comment.

Point 3: The abstract should state briefly the purpose of the research, the principal results and major conclusions. An abstract is often presented separately from the article, so it must be able to stand alone

Response: Required changes have been made in the abstract.

Introduction

Point 4: Discuss the research aims, research gap and discuss the paper layout Add up-to-date references to support your discussion

Response: The comment has been addressed in the introduction.

Point 5: The necessity and innovation of the article should be presented to the introduction

Response: Comment addressed in the introduction

Methods and Materials 

Point 6: The methodology of this study should be detailed, limit information was provided on method and materials

Response: Comment addressed in the methods part.

Point 7: The authors should be able to control/reduce the selection bias

Response: The selection bias was controlled by recruiting the controls from the same population as the cases and by matching. Comment addressed under the participant's recruitment.

Point 8: Abbreviations should be defined

Response: Comment addressed.

Discussion

Point 9: I believe that more in depth discussion is needed. The discussion as present now is simple and concise. Revision of more papers using similar technique is needed

Response: Comment addressed.

Point 10: In the discussion, please discuss if the study research questions are answered or not Also introduce the model in detail. Draw a conclusion from this study and present the limitations and future research.

Response: Comment addressed.

Reviewer 2 Report

Please address the following points:

- The introduction is highly repetitive. Please structure the information related to Pakistan, presented firstly at the national level and then within the regional specificities. This should provide a better contextual understanding of the deep-rooted causes for failure to vaccinate.

- In the introduction please also comment succinctly on the relative numbers of cases in Afghanistan. This would provide a comparator to the more detailed data that you are correctly providing for Pakistan.

- In the introduction you state "which are a significant cause of death amongst children, especially within resource-constrained regions of the country", that is absolutely correct, please provide a reference or an indicative number from Pakistan, to aid the contextual understanding.

- In the introduction you state "To design a strategy to overcome these challenges, we must have thorough knowledge and comprehension of the various social and cultural reasons underlying vaccination refusals within specific areas". That is correct, however you already provided a long list of general challenges - as such please ensure that the knowledge gaps are highlighted better, e.g., for those specific population groups.

- In the methods, were participants excluded if questionnaires were incomplete?

- The discussion follows the results faithfully and is clearly presented. However, it does not include any information that is region specific. For example you state that "the burden of traveling on foot with one or more children, has the strong potential to deter people from seeking and returning for vaccination services" - that is absolutely fine, but there is no statement regarding the density of healthcare station coverage. 

- Similarly, you state "likely due to the outreach efforts of health workers and vaccinators", but you provide no specific examples for when those have taken place

Author Response

Point 1: The introduction is highly repetitive. Please structure the information related to Pakistan, presented firstly at the national level and then within the regional specificities. This should provide a better contextual understanding of the deep-rooted causes for failure to vaccinate.

Response: Comment addressed

Point 2: In the introduction please also comment succinctly on the relative numbers of cases in Afghanistan. This would provide a comparator to the more detailed data that you are correctly providing for Pakistan.

Response: As per the national survey, the vaccination uptake in Afghanistan has been added to the introduction.

Point 3: In the introduction you state "which are a significant cause of death amongst children, especially within resource-constrained regions of the country", that is absolutely correct, please provide a reference or an indicative number from Pakistan, to aid the contextual understanding.

Response: The introduction has been revised, and references are added where required.

Point 4: In the introduction you state "To design a strategy to overcome these challenges, we must have thorough knowledge and comprehension of the various social and cultural reasons underlying vaccination refusals within specific areas". That is correct, however you already provided a long list of general challenges - as such please ensure that the knowledge gaps are highlighted better, e.g., for those specific population groups.

Response: Comment addressed

Point 5: In the methods, were participants excluded if questionnaires were incomplete?

Response: The study had a 100% response rate, and all the participants completed the questionnaire. Hence no one was excluded.

Point 6: The discussion follows the results faithfully and is clearly presented. However, it does not include any information that is region specific. For example you state that "the burden of traveling on foot with one or more children, has the strong potential to deter people from seeking and returning for vaccination services" - that is absolutely fine, but there is no statement regarding the density of healthcare station coverage. 

Response: The discussion has been revised and presented in detail

Point 7: Similarly, you state "likely due to the outreach efforts of health workers and vaccinators", but you provide no specific examples for when those have taken place

Response: The discussion has been revised and presented in detail

Reviewer 3 Report

Manuscript (ID: vaccines-2357877) presented results of matched case-control study about factors associated with vaccine refusal (polio and routine immunization) in high-risk areas of Karachi, Pakistan. But, some corrections are needed (major revision): 

  • Line 18: Instead of `refusals for OPV/IPV', enter `refusals for OPV/IPV in campaigns (NIDs and SIAs)'. Instead of `RI` write `routine immunization (RI)`.    

  • Lines 39-89: In the Introduction section, data on the epidemiological situation of wild poliomyelitis are presented, with special reference to the polio endemic country - Pakistan. Also, numerous problems with the implementation of the global eradication program of poliomyelitis were emphasized, and especially in connection with the difficulties in the implementation of vaccination against poliomyelitis during the final phase of eradication in endemic countries.    

  • Lines 84-85: Cite the appropriate reference in this sentence. Define the term `essential vaccines', and list those vaccines in this sentence.   

  • Lines 88-91: Align the stated objectives of this study with the text on Lines 93-97. It is a good practice to state the objectives of the study at the end of the Introduction section, which is done in this manuscript. However, it is not a good practice to state the objective in the Methods section. Harmonize those two parts of the manuscript text and avoid unnecessary repetition in writing.  

  • Line 108: The `target population' is specified. Specify for this study inclusion criteria and exclusion criteria. Specify the type of study sample. State the method of recruiting participants for this study. Specify `Participation rate` and `Response rate`.  

  • Lines 108-109: When the specified study was conducted (previous Maternal and Child Care Program (MCCP) phase 2 study), cite the appropriate reference.  

  • Lines 112-114: In this sentence, 3 groups of cases are clearly defined. In the further part of the work (Methods, Results, Discussion, Conclusion), it is very important for the readers of this work that the authors completely and consistently adhere to the above definitions. Any shortening of the description of these 3 groups may lead to unnecessary ambiguities and confusion in the interpretation of the results.   

  • Lines 131-134: In section METHODS (subsection Data Collection) define all described variables (as well as the classification of those variables): Knowledge level, SES Level, etc. 

  • Line 146: List all variables for which adjustment was made.   

  • Lines 160-161: Explain the term `significant' in this sentence, since Table 1 does not show the results of any statistical significance test. Or replace with an appropriate term.  

  • Line 189: Explain exactly the meaning of `*Group A:' as `Group A: Refusal for routine immunization (RI) but acceptance of OPV'. Corrections entered for Tables 1-3.    

  • Line 190: Explain exactly the meaning of `*Group B:` as `Refusal of OPV (national immunization days (NIDs) & SIAs) but acceptance of RI`. Corrections entered for Tables 1-3. 

  • Line 211: Instead of `Group A: Routine immunization refusals' write `Refusal for routine immunization (RI) but acceptance of OPV'.  

  • Line 230: Instead of `Group B: Oral Polio Vaccine refusal in campaigns (NIDs and SIAs)' write `Refusal of OPV (national immunization days (NIDs) & SIAs) but acceptance of RI'.   

  • Line 262: Reconstruct the Discussion section in a way to ensure a logical flow in the presentation of all significant results of this study.  

  • Lines 262-289: The references in this text are not listed in order. Align the order with the list of References.  

  • Lines 263-293: In the Discussion section, only 4 references are cited, whereby 2 and 3 references refer to Pakistan. In the discussion, it is mandatory to compare the results of this study with the results of similar studies in other countries. Give possible explanations for the described differences in results in different studies, citing appropriate references.  

  • Line 293: Add a new paragraph in which `Strength and Limitations of the study' will be discussed in detail. 

  • Lines 297-303: Unnecessary repetition of ALL results from this study in the conclusion. Correct this.  

The quality of the English language is acceptable.     

Author Response

Point 1: Line 18: Instead of `refusals for OPV/IPV', enter `refusals for OPV/IPV in campaigns (NIDs and SIAs)'. Instead of `RI` write `routine immunization (RI)`.    

Response: Comment addressed. Abbreviations have been expanded.

Point 2: Lines 39-89: In the Introduction section, data on the epidemiological situation of wild poliomyelitis are presented, with special reference to the polio endemic country - Pakistan. Also, numerous problems with the implementation of the global eradication program of poliomyelitis were emphasized, and especially in connection with the difficulties in the implementation of vaccination against poliomyelitis during the final phase of eradication in endemic countries.    

Response: Acknowledged.

Point 3: Lines 84-85: Cite the appropriate reference in this sentence. Define the term `essential vaccines', and list those vaccines in this sentence.   

Response: Comment addressed

Point 4: Lines 88-91: Align the stated objectives of this study with the text on Lines 93-97. It is a good practice to state the objectives of the study at the end of the Introduction section, which is done in this manuscript. However, it is not a good practice to state the objective in the Methods section. Harmonize those two parts of the manuscript text and avoid unnecessary repetition in writing.  

Response: Comment addressed

Point 5: Line 108: The `target population' is specified. Specify for this study inclusion criteria and exclusion criteria. Specify the type of study sample. State the method of recruiting participants for this study. Specify `Participation rate` and `Response rate`.  

Response: Some details are already present in the methods, while the missing information has been added as advised.

Point 6: Lines 108-109: When the specified study was conducted (previous Maternal and Child Care Program (MCCP) phase 2 study), cite the appropriate reference.  

Response: Reference added

Point 7: Lines 112-114: In this sentence, 3 groups of cases are clearly defined. In the further part of the work (Methods, Results, Discussion, Conclusion), it is very important for the readers of this work that the authors completely and consistently adhere to the above definitions. Any shortening of the description of these 3 groups may lead to unnecessary ambiguities and confusion in the interpretation of the results.   

Response: Acknowledged.

Point 8: Lines 131-134: In section METHODS (subsection Data Collection) define all described variables (as well as the classification of those variables): Knowledge level, SES Level, etc. 

Response: Details added to the methods part.

Point 9: Line 146: List all variables for which adjustment was made.   

Response: Since we were interested in finding the factors/barriers associated with each refusal category, the model only included significant variables. Hence no adjustment was necessary.

Point 10: Lines 160-161: Explain the term `significant' in this sentence, since Table 1 does not show the results of any statistical significance test. Or replace with an appropriate term.  

Response: Acknowledged. The word has been replaced with an appropriate one.

Point 11: Line 189: Explain exactly the meaning of `*Group A:' as `Group A: Refusal for routine immunization (RI) but acceptance of OPV'. Corrections entered for Tables 1-3.    

Response: Comment addressed.

Point 12: Line 190: Explain exactly the meaning of `*Group B:` as `Refusal of OPV (national immunization days (NIDs) & SIAs) but acceptance of RI`. Corrections entered for Tables 1-3. 

Response: Comment addressed

Point 13: Line 211: Instead of `Group A: Routine immunization refusals' write `Refusal for routine immunization (RI) but acceptance of OPV'.  

Response: Comment addressed

Point 14: Line 230: Instead of `Group B: Oral Polio Vaccine refusal in campaigns (NIDs and SIAs)' write `Refusal of OPV (national immunization days (NIDs) & SIAs) but acceptance of RI'.   

Response: Comment addressed

Point 15: Line 262: Reconstruct the Discussion section in a way to ensure a logical flow in the presentation of all significant results of this study.  

Response: Comment addressed

Point 16: Lines 262-289: The references in this text are not listed in order. Align the order with the list of References.  

Response: The discussion part has been revised.

Point 17: Lines 263-293: In the Discussion section, only 4 references are cited, whereby 2 and 3 references refer to Pakistan. In the discussion, it is mandatory to compare the results of this study with the results of similar studies in other countries. Give possible explanations for the described differences in results in different studies, citing appropriate references.  

Response: The discussion part has been revised.

Point 18: Line 293: Add a new paragraph in which `Strength and Limitations of the study' will be discussed in detail. 

Response: Comment addressed

Point 19: Lines 297-303: Unnecessary repetition of ALL results from this study in the conclusion. Correct this.  

Response: Comment addressed

Round 2

Reviewer 3 Report

Thank you for the opportunity to re-review manuscript ID: vaccines-2357877. In the revised version of this manuscript, the authors insert the significant changes and satisfactorily addressed all of my comments.  

  • Corrections in the Abstract made this part of the manuscript clearer.
  • The Introduction section is satisfactorily supplemented with data on poliomyelitis vaccination in Pakistan.
  • The section Methods substantially was corrected in revised version of this manuscript (including the appropriate data for eligibility cases, inclusion and exclusion criteria, data collection, questionnaires used, etc).
  • In revised version, the Discussion section is substantially corrected, and now with the logical flow of the presentation provides comparison of own results with results of similar researches.
  • In the revised version of this manuscript, the new paragraph, `Limitations of the study` was added.
  • Conclusions are clear.  

The authors have done a good job. I thank the authors.   

Quality of English language is appropriate.